# Understanding the Half-Life Extension of Intravitreally Administered Antibodies Binding to Ocular Albumin

**DOI:** 10.3390/pharmaceutics12090810

**Published:** 2020-08-26

**Authors:** Simon Hauri, Paulina Jakubiak, Matthias Fueth, Stefan Dengl, Sara Belli, Rubén Alvarez-Sánchez, Antonello Caruso

**Affiliations:** 1Roche Pharma Research and Early Development, Pharmaceutical Sciences, Roche Innovation Center Basel, F. Hoffmann-La Roche Ltd., Grenzacherstrasse 124, CH-4070 Basel, Switzerland; jakubiak.paulina@gene.com (P.J.); matthias.fueth@roche.com (M.F.); sara.belli@roche.com (S.B.); ruben.alvarez_sanchez@roche.com (R.A.-S.); antonello.caruso@roche.com (A.C.); 2Roche Pharma Research and Early Development, Large Molecule Research, Roche Innovation Center Munich, F. Hoffmann-La Roche Ltd., Nonnenwald 2, D-82377 Penzberg, Germany; stefan.dengl@roche.com

**Keywords:** ocular drug delivery, ocular pharmacokinetics, half-life extension, albumin, therapeutic proteins, size exclusion chromatography

## Abstract

The burden associated with frequent injections of current intravitreal (IVT) therapeutics may be reduced by long-acting delivery strategies. Binding to serum albumin has been shown to extend the ocular half-life in rabbits, however, the underlying molecular mechanisms and translational relevance remain unclear. The aim of this work was to characterize the in vitro and in vivo formation of complexes between human serum albumin (HSA) and an antigen-binding fragment of a rabbit antibody linked to an anti-HSA nanobody (FabA). The ocular and systemic pharmacokinetics of ^3^H-labeled FabA (0.05 mg/eye IVT) co-formulated with HSA (1 and 15 nmol/eye) were assessed in Dutch belted rabbits. Next, FabA was incubated in vitreous samples from cynomolgus monkeys and human donors (healthy and diseased) supplemented with species-specific serum albumin. Finally, the FabA-albumin complexes formed in vitro and in vivo were analyzed by radio-size exclusion chromatography. A 3-fold increase in FabA vitreal exposure and half-life was observed in rabbits co-administered with 15 nmol HSA compared to 1 nmol and a control arm. The different pharmacokinetic behavior was explained with the formation of higher molecular weight FabA–albumin complexes. The analysis of vitreous samples revealed the existence of predominantly 1:1 complexes at endogenous or low concentrations of supplemented albumin. A shift towards 1:2 complexes was observed with increasing albumin concentrations. Overall, these results suggest that endogenous vitreal albumin concentrations are insufficient for half-life extension and warrant supplementation in the dosing formulation.

## 1. Introduction

As the number of patients with vision-threatening conditions has been steadily increasing in recent years, largely driven by the growth and ageing of the world’s population [1], so have the efforts directed at finding novel treatments. As a notable example, the development of anti-vascular endothelial growth factor (VEGF) therapeutics has transformed the management of neovascular age-related macular degeneration (nAMD), diabetic retinopathy (DR) and diabetic macular edema (DME) [2,3]. The antibodies ranibizumab (Lucentis^®^) and aflibercept (Eylea^®^) are the current standard of care to treat these retinal diseases [4,5], with off-label use of bevacizumab (Avastin^®^) being common [4]. Recently, a novel fusion protein, brolucizumab (Beovu^®^), was also approved for the treatment of nAMD [6,7]. All anti-VEGF biologics are administered through intravitreal (IVT) injections every 4–12 weeks [8,9]. The dosing frequency can be reduced in clinical practice, leading in some cases to suboptimal treatment [10]. In addition, IVT injections are invasive and can pose a significant medical, psychological, and financial burden on patients and healthcare systems alike. For these reasons, treatment options with less frequent dosing regimens are desirable, potentially improving outcomes and compliance [11].

To address such needs, a common approach in protein drug discovery is to optimize the molecular properties influencing the target suppression and maximum feasible dose, such as the binding affinity, stability and aggregation. In addition, durability can be enhanced by reducing the rate of ocular elimination, which is driven by diffusion from the vitreous body to the anterior chamber and by aqueous humor (AH) turnover [12]. Diffusivity decreases inversely with macromolecular size, so larger antibody formats, PEGylation or conjugation to biopolymers result in prolonged ocular retention [13,14,15,16,17,18]. This may also be achieved with binding to constituents of the vitreous matrix, such as collagen (type II) and hyaluronic acid [19,20], or association with soluble proteins, like albumin [21]. Parallel to the development of novel therapeutic agents, continuous effort is being put into designing long-acting delivery strategies applicable to existing and new drugs [22,23,24]. For instance, the Port Delivery System, an implantable intraocular device, is currently undergoing clinical investigation for sustained delivery of ranibizumab on a 6-month refill regimen [25].

Among these approaches, binding to albumin may be considered appealing because of its undetermined biological function [26] and relatively high abundance in ocular fluids. In the human vitreous, albumin was shown to be approximately equimolar to the standard-of-care doses of IVT antibodies [20]. However, its small size (66.4 kDa) and correspondingly short ocular half-life (reported as 2–4 days in rabbit [27,28]) may limit the prolongation of ocular half-life and therefore its suitability as a retention target.

The effects of albumin binding on antibody pharmacokinetics (PK) were recently investigated in rat and rabbit eyes by Fuchs and Igney [21]. To this end, a nanobody molecule (BI-X, 40 kDa) was used containing one high affinity binding site to human serum albumin (HSA) and showing no specific binding to rat or rabbit albumin. In rabbits, co-administration with albumin led to a 3-fold increase in the vitreal half-life of BI-X (10.1 days with HSA vs. 3.2 days without). Association of one BI-X and one albumin molecule would result in a 106 kDa complex, however, the observed half-life exceeded that of macromolecules with similar molecular weight (MW) (40–150 kDa, 2–6 days [29,30]). Fuchs and Igney [21] hypothesized complexation with multiple HSA molecules, since commercially available human and bovine serum albumins are known to form oligomers after reconstitution from powders [31,32]. While oligomeric complexes could theoretically explain the long retention observed in vivo, their nature and stoichiometry were not investigated experimentally in the original study [21]. It also remains unclear whether the endogenous albumin concentration in the human vitreous is conducive to a meaningful half-life extension.

In this work, we investigated the molecular mechanisms underlying the binding of macromolecules to vitreal albumin by means of in vitro and in vivo experiments. We aimed to determine the vitreal albumin concentration leading to the formation of high MW complexes and understand the potential impact on ocular drug retention. To this end, we assessed the IVT PK in rabbits of a radiolabeled antigen-binding mAb fragment (Fab) linked to an anti-albumin nanobody and co-formulated with different amounts of albumin. We incubated the Fab in cynomolgus monkey and human vitreous humor (VH), supplemented with different amounts of species-specific serum albumin. Finally, we characterized the nature of Fab–albumin complexes formed in vitro and in vivo with radio-size exclusion chromatography (radio-SEC). Herein, we present the results of these investigations and assess the scope and translatability of albumin binding as a long-acting drug delivery approach.

## 2. Materials and Methods

### 2.1. Materials

A HSA binding immunoglobulin single variable domain (nanobody, VHH) sequence (ALB8 from WO 2012/131078 A1 [33]) was recombinantly fused via a 3× GGGGS-linker to the C-terminus of the heavy chain of a species-matched rabbit Fab (rabFab) [17]. The C-terminus of the nanobody was modified by a His6-tag (FabA). In addition, rabFab with a 3× GGGGS-linker-His6-tag at the C-terminus of the heavy chain (without the nanobody sequence) was produced as the non-albumin binding control molecule (FabB). The MW of the antibodies was 59.5 and 47.4 kDa for FabA and FabB, respectively.

The DNA sequences were cloned into mammalian expression vectors that were used for transfection of HEK293F cells (Invitrogen, Waltham, MA, USA). After expression of the proteins, cleared supernatants were subjected to affinity chromatography (cOmpleteTM His-Tag purification resin, Sigma-Aldrich, St Louis, MO, USA). Bound protein was eluted in 50 mM Na_2_HPO_4_ (Sigma-Aldrich, St Louis, MO, USA), 300 mM NaCl (Sigma-Aldrich, St Louis, MO, USA), 500 mM imidazole (Sigma-Aldrich, St Louis, MO, USA), pH 8.0. Eluted protein was subjected to preparative size exclusion chromatography (SEC) (HiLoad 26/60 Superdex 200 prep grade, GE Healthcare Life Sciences, Marlborough, MA, USA) in 1× PBS, pH 7.0. The biochemical quality of the final product was high with a 100% monomer peak on analytical high-performance liquid chromatography (HPLC) SEC (BioSuite, Waters Corporation, Milford, MA, USA) for both proteins. Identity of the proteins was verified by mass spectrometry.

Human, rabbit (RSA) and porcine serum albumin (PSA) were purchased from Sigma-Aldrich (St Louis, MO, USA). Cynomolgus serum albumin (CSA) was purchased from antibodies-online GmbH (Aachen, Germany) and Abcam (Cambridge, UK). Frozen cynomolgus monkey and human (diabetic with and without retinopathy) VH samples were provided by Anawa Trading SA (Kloten, Switzerland).

### 2.2. Albumin Binding Kinetics and Cross-Reactivity

In order to determine the affinity of the Fabs to serum albumin from different species, interaction of the compounds with HSA, CSA, RSA, and PSA was tested by surface plasmon resonance. The selected albumin concentration range was based on the binding affinity of the nanobody for HSA reported in the original patent [33].

In brief, an anti-His capturing antibody (28995056, GE Healthcare Life Sciences, Marlborough, MA, USA) was immobilized to a Series S Sensor Chip CM3 (29104990, GE Healthcare Life Sciences, Marlborough, MA, USA) using standard amine coupling chemistry, resulting in a surface density of approximately 5000 resonance units. As the running and dilution buffer, HBS-P+ (10 mM HEPES (Sigma-Aldrich, St Louis, MO, USA), 150 mM NaCl pH 7.4, 0.05% surfactant P20 (Sigma-Aldrich, St Louis, MO, USA)) was used. The measurement temperature was set to 25 °C. The antibodies were captured to the surface with resulting capture levels of approximately 15 resonance units. Dilution series of HSA, CSA, RSA, and PSA in the range 22.2–1800 nM were injected for 180 s, dissociation was monitored for 300 s at a flow rate of 30 µL/min. Subsequently, the surface was regenerated by injecting 10 mM glycine pH 1.5 for 60 s. Bulk refractive index differences were corrected by subtracting blank injections and by subtracting the response obtained from the control flow cell without captured antibody. Rate constants were calculated using the Langmuir 1:1 binding model within the Biacore Evaluation software (GE Healthcare Life Sciences, Marlborough, MA, USA).

### 2.3. Native Intact Mass Spectrometry of Protein Complexes

FabA–HSA complex composition was characterized by high-resolution intact mass spectrometry. Protein separation was carried out by SEC on an Ultimate 3000 UHPLC system (Thermo Fisher Scientific, Waltham, MA, USA) using an Acquity BEH SEC column (200 Å, 1.7 × 100 mm; Waters Corporation, Milford, MA, USA) with 100 mM ammonium acetate pH 6.8 as mobile phase. The method run time was 10 min at 250 µL/min isocratic flow. Intact mass analysis was performed on a maXis II qToF instrument (Bruker Corporation, Billerica, MA, USA). The ion polarity was set to positive mode, covering the mass range of 800–10,000 *m*/*z*. Spectra were recorded at 0.5 Hz. Capillary voltage was 4500 V at a nebulizer pressure of 2 bar and a dry gas flow rate of 10 L/min at 280 °C. In-source fragmentation was enabled at 30 eV. The raw files were analyzed using Compass DataAnalysis 4.4 SR1 (Bruker Corporation, Billerica, MA, USA) and PMI-Intact v3.8-11 x64 (Protein Metrics Inc., San Carlos, CA, USA).

### 2.4. Radiolabeling

FabA and FabB were chemically labeled with ^3^H-propionic acid esterification activated by N-hydroxysuccinimide [34]. The specific activity of FabA was 1.5 mCi/mg (90 Ci/mmol) and yielded a protein concentration of 2.15 mg/mL. The specific activity of FabB was 2.3 mCi/mg (110 Ci/mmol) and yielded a protein concentration of 2.23 mg/mL. The endotoxin levels measured after sterile filtration using 0.20 µm syringe filter-discs were 0.832 EU/mL and 0.641 EU/mL for FabA and FabB, respectively.

### 2.5. Ocular Pharmacokinetics Study in Rabbits

An in vivo study was performed with the goal of investigating the effects of albumin binding on ocular pharmacokinetics. The animal experimentation was approved by the local veterinary authorities and was conducted in strict accordance with the Swiss federal regulations on animal care and laboratory use and in adherence to the rules of the Association for Assessment and Accreditation of Laboratory Animal Care International. The husbandry license number is 1009H F. Hoffmann-La Roche Ltd. and animal permission number 2764 (valid since 01.07.2018).

^3^H-labeled FabA and FabB were administered to female Dutch belted rabbits allocated to three study groups (*n* = 4/group). Each group received 0.05 mg Fab premixed with HSA. Group 1 was dosed FabA with 1 nmol HSA, group 2 FabA with 15 nmol HSA, and group 3 FabB with 1 nmol HSA. Molar quantities of HSA were selected based on serum albumin concentration values reported in healthy humans and cynomolgus monkeys (group 1 and 3) as well as patients with different types of retinopathies (group 2). Table 1 summarizes all values available in the literature to date, to the best of our knowledge. A volume of 50 µL of the filtered (0.22 µm) formulation was injected intravitreally with a syringe (BD Micro-Fine, 30 G × 8 mm) to both eyes. Blood was withdrawn from the ear vein into K3-EDTA tubes and plasma produced by centrifugation (10 min at 2000× *g* at 4 °C) at 0, 1, 2, 5, 7, 24, 48, 72, 168, and 336 h post-dose. AH and VH samples were collected in-life from anesthetized rabbits at 24, 74, 168, and 336 h post-dose in a composite design. For AH sampling, a 30 G needle was inserted through the cornea into the anterior chamber and a maximum of 20 µL of aqueous was sampled. For VH sampling, a 22 G butterfly needle was inserted through the sclera and pars plana for 5–10 mm, and a maximum of 20 µL of vitreous was gently aspirated from the center of the vitreal cavity. Safety of the method and possible effects on the eye were evaluated by frequent ophthalmic examinations (biomicroscopy and indirect ophthalmoscopy) over the study period. Radioactivity in the VH, AH and plasma samples was counted with a TopCount NXT HTS Microplate Counter (Perkin Elmer, Waltham, MA, USA). A non-compartmental analysis of the composite concentration-time profiles was performed in Phoenix WinNonlin version 6.4 (Certara USA Inc., Princeton, NJ, USA, 2015).

### 2.6. In Vitro Sample Preparation

To test the contribution of albumin to multimeric complex formation, FabA was incubated in vitro with different concentrations of HSA and CSA in PBS, human and cynomolgus monkey vitreous humor (VH) at 37 °C for 18 h, as summarized in Table 2. Low (0.57 µM) and high (5.7 µM) dose levels of radiolabeled FabA were tested. For the high dose, the specific activity of FabA was kept constant and supplemented with unlabeled FabA. The samples were stored at −20 °C prior to further analysis. Cynomolgus monkey and human VH samples collected post mortem were obtained from a commercial provider. Upon receipt of the samples, albumin concentration was determined using the Millipore Human ALB/Serum albumin ELISA kit (Sigma-Aldrich Chemie GmbH, Buchs, Switzerland) according to manufacturer’s instructions. Absorbance at 450 nm was measured on an EnSpire Multimode Plate Reader (Perkin Elmer, Waltham, MA, USA).

### 2.7. Radio Size Exclusion Chromatography

To characterize the nature of Fab–albumin complex formation in vivo, SEC studies were performed with the vitreous humor samples obtained in the rabbit study. As outlined in Table 2, FabA and FabB were also incubated with HSA and CSA in PBS, as well as in human and cynomolgus monkey VH at 37 °C for 18 h. The samples were stored at −20 °C prior to SEC analysis.

Protein separation was carried out by SEC on an Agilent 1290 Infinity II UHPLC system (Agilent Technologies Inc, Santa Clara, CA, USA) using an Acquity BEH SEC column (200 Å, 1.7 × 100 mm, Waters Corporation, Milford, MA, USA) with 100 mM sodium phosphate buffer at pH 7.4 containing 10% ethanol as mobile phase. The method run time was 10 min at 250 µL/min isocratic flow. Fractions were collected between 4 and 10 min for 0.05 min (12.5 µL) in a Deepwell LumaPlate384 (Perkin Elmer, Waltham, MA, USA) and radioactivity was measured on a TopCount NXT HTS Microplate Counter (Perkin Elmer, Waltham, MA, USA).

## 3. Results

### 3.1. Albumin Binding Kinetics Experiments

FabA and FabB binding kinetics to the serum albumin of different species are summarized in Table 3. FabA showed a high affinity interaction with HSA in a two-digit nanomolar range. The molecule was fully cross-reactive to CSA with comparable on- and off-rates, but did not interact with RSA or PSA. For FabB, no interaction with serum albumin from the different species was detected.

Verification of HSA binding after radiolabeling of the compounds for use in vivo confirmed that the labeling procedure did not have an impact on their binding properties (Table 4).

### 3.2. Native LC-MS

To investigate the composition of the protein complexes, unlabeled FabA was incubated with HSA and analyzed by size exclusion chromatography coupled to mass spectrometry (SEC-MS). The experiment was performed under non-denaturing conditions at neutral pH to prevent protein complex dissociation during the chromatographic separation. By measuring the intact mass of FabA, albumin and the FabA–albumin complexes, the composition of the different molecular weight species was investigated (Figure 1). The UV chromatogram shows four distinct peaks at 280 nm absorption (Figure 1A). Fifteen µM of FabA and HSA were measured separately, to define migration times and reference mass spectra (Figure 1A, peak 1 and 2). To favor 1:1 complex formation, 15 µM of FabA was incubated with 15 µM of HSA, resulting in a 126 kDa complex at a migration time of 4.8 min (Figure 1A, peak 3). For the formation of the 1:2 complexes, 15 µM FabA and 75 µM HSA were incubated and detected at a migration of time 4.2 min with masses for FabA and multimeric albumin (Figure 1A, peak 4). The underlying mass spectra in all four peaks were deconvoluted to intact protein masses. Peak 1 consisted of a protein with the mass of 59 kDa, corresponding to FabA (Figure 1B). It is also in good alignment with the molecular weight standard with a faster migration time than the 66 kDa marker. Peak 2 contained a single mass of 67 kDa, which is the mass of HSA (Figure 1C). Migration time of peak 3 was just before the 150 kDa marker and when deconvoluted, several masses were identified: The 59 kDa mass of FabA, the 67 kDa of HSA and a 126 kDa mass corresponding to the 1:1 FabA–HSA complex (Figure 1D). The slowest migrating peak 4 contained again FabA, but no 67 kDa HSA. Instead, HSA was found in its dimeric form at 133 kDa. The 126 kDa complex was still present and additionally two large masses were found at 186 and 193 kDa (Figure 1E). These high molecular weight complexes could consist of two FabA and one HSA or vice versa.

### 3.3. Ocular Pharmacokinetics Study

Ocular and plasma PK were studied in rabbits after an IVT injection of human albumin-binding FabA (0.05 mg; 0.84 nmol; 50 µL) co-formulated with HSA at molar quantities of 1 (group 1) and 15 nmol (group 2). An equal dose of FabB lacking the albumin-binding moiety was co-administered with 1 nmol HSA and served as negative control (group 3). At each time point, VH, AH and plasma concentrations of the test molecules were calculated as mean ± standard deviation (S.D.) from four individual samples. The composite concentration-time profiles in rabbit ocular fluids and plasma are depicted in Figure 2, and the individual data is reported in the Appendix A. The pharmacokinetic parameters calculated by non-compartmental analysis are summarized in Table 5.

The VH PK data demonstrated that approximately a 3-fold extension of vitreous half-life was achieved in the group that received FabA with the excess of 15 nmol HSA (*t*_1/2_ = 149 h) when compared to FabB dosed with 1 nmol HSA (*t*_1/2_ = 52 h) or FabA co-administered with an equimolar amount of HSA (*t*_1/2_ = 58 h). The observed prolongation of the vitreous half-life, related to a decrease in the ocular clearance, resulted in an increased overall exposure to FabA in the VH in low and high HSA concentration groups (AUCinf = 104 and 230 h∙nmol/mL, respectively) compared to the FabB group (AUCinf = 72.8 h∙nmol/mL). Similarly, longer residence times were observed for HSA-binding FabA (MRTinf = 79.1 h for group 1 and 129 h for group 2) than for FabB (MRTinf = 68.6 h). The volume of distribution based on nominal dose was calculated as 0.68, 0.79 and 0.87 mL for groups 1, 2 and 3, respectively.

The concentration measurements in AH indicated a later *t*_max_ for FabA (168 h for 1 nmol HSA and 74 h for 15 nmol HSA) than FabB (24 h). Minimal differences were observed in the AH exposure across groups, with the AUCinf calculated to be 5.32, 6.68 and 4.70 h∙nmol/mL for group 1, 2 and 3, respectively. These comparable AUC values could be explained by a complete transfer of the compounds from the vitreous into the anterior chamber and a comparable anterior clearance across molecules, dictated by the AH flow. The respective *t*_1/2_ values in the aqueous humor were calculated to 50.9, 87.0 and 50.4 h.

Both FabA and FabB were found circulating in plasma at very low levels with similar exposures across study groups, in line with the lack of cross-reactivity with rabbit serum albumin. Nonetheless, the average plasma concentration of FabA in both low and high HSA concentration groups showed a bi-phasic behavior in the systemic absorption phase, with a first lower concentration peak and a second absorption peak appearing later than the FabB *t*_max_. This effect was more prominent and consistently observed in all individuals of the 15 nmol HSA group, possibly reflecting differences in ocular kinetics and systemic absorption rates between the free and complexed FabA.

### 3.4. Radio Size Exclusion Chromatography Studies on Rabbit Vitreous Samples and Dosing Formulations

To investigate the underlying molecular mechanism of half-life extension in vivo, the rabbit VH samples were analyzed by SEC and radiometric detection. SEC allows to determine the stoichiometry of the FabA–albumin complexes by separating them by their hydrodynamic radius. For the VH samples from the group dosed at 1:1 ratio of FabA and HSA, the radio-signal for FabA was observed at a retention time corresponding to a MW between 66 and 150 kDa. This was in good agreement with the theoretical MW of the expected 1:1 complex, namely 126 kDa (Figure 3A). For the VH samples of the 15:1 molar ratio group, the radio signal shifted to the higher MW species greater than 150 kDa, consistent with the MW of a 1:2 complex of about 180 kDa (Figure 3B).

These findings suggest the formation of higher stoichiometry complexes because of the higher albumin concentration that could be responsible for the ocular half-life extension observed in vivo.

SEC analysis of the dosing formulations was also performed in order to verify whether the multimeric complex is formed prior to ocular administration (Figure 4). A concentration range of HSA was tested with FabA to determine a stoichiometric dependency. Additionally, FabA was tested for complex formation with CSA. In the absence of albumin, FabA migrated slightly faster than the 66 kDa marker, as expected from its MW of 59.5 kDa (Figure 4A). Under equimolar conditions, both HSA and CSA predominantly formed the 126 kDa 1:1 complex (Figure 4B,C). In the presence of a 15-fold molar excess of albumin, higher MW complexes were formed by both albumin species variants.

These data indicate that the half-life extending 1:2 stoichiometry can be induced already by co-formulation prior to administration.

### 3.5. FabA–Albumin Complex Formation in Monkey and Human Vitreous Humor

Given the lack of rabbit cross-reactivity, co-administration of HSA was necessary in the PK study. To assess the complexation profile under more physiological conditions and build the rationale for a potential in vivo study in non-human primates, cynomolgus monkey and human VH samples were tested in vitro for Fab–albumin complex formation.

A VH sample from a cynomolgus monkey was incubated with FabA and supplemented with increasing concentrations of CSA to saturate FabA binding (Figure 5). At endogenous albumin levels, a 1:1 stoichiometry complex could be observed. Upon supplementation with 50, 100 and 250 µM CSA, a shift towards higher MW species was apparent. Increasing the concentration of FabA by 10-fold had a limited effect on the MW of the complexes. The endogenous cynomolgus monkey albumin concentration was determined by ELISA to be 35 mg/L (0.53 µM) (Table 1). This low basal albumin concentration relative to FabA explains why, in the absence of CSA supplementation, only a mixture of non-complexed FabA and 1:1 complexes was detected by SEC.

In a human post mortem vitreous sample from a single donor without eye disease, the endogenous albumin was not conducive to the formation of the 1:2 complex (Figure 6). When exogenous HSA was added, a shift towards the 1:2 stoichiometry could be clearly observed at 50 µM HSA. Increasing the FabA concentration by 10-fold, most FabA remained unbound at endogenous albumin concentrations and a progressive shift to the 1:1 and 1:2 complex was observed with HSA supplementation. The endogenous albumin concentration was determined to be 280 mg/L, or 4.22 µM (Table 1), a level that appears insufficient to form higher MW complexes.

A human post mortem VH sample from a single donor with diabetic retinopathy was tested in similar conditions (Figure 7). In the absence of albumin supplementation, mostly 1:1 complexed FabA was detected. Increasing the albumin concentration by supplementation of 50 µM resulted in the formation of the 1:2 complex. The albumin concentration was measured as 699 mg/L (10.5 µM), consistent with previous reports [35,36,38,39] (Table 1).

## 4. Discussion

Prolonging the duration of ocular exposure and target engagement is one of the main goals in the optimization of intravitreal antibodies. Most approaches currently pursued for ocular half-life extension aim to reduce the drug’s diffusivity. This may be accomplished by binding to a vitreous matrix constituent, such as hyaluronic acid or collagen [19,20], by designing a bulky format of large hydrodynamic radius [14,15,16,17] or by enabling it to associate with soluble endogenous macromolecules, such as albumin, and form high MW complexes.

Previously, Fuchs and Igney [21] demonstrated in rats and rabbits that binding to vitreal albumin produces a meaningful increase in ocular retention using an experimental antibody. In this work, we assessed the scope and translatability of this approach by means of a series of in vitro and in vivo investigations with a rabbit Fab linked to an anti-HSA nanobody, FabA (Table 2).

By means of SEC coupled to UV and MS detection, we first confirmed the identity of both HSA and FabA (Figure 1B,C). Next, we showed that following incubation at equimolar concentrations, a new entity was formed with a retention time and mass consistent with a 1:1 complex (Figure 1D). Last, in conditions of excess of albumin, an additional chromatographic peak consistent with a 1:2 stoichiometry was observed. Given the similar MW of the two possible 1:2 complexes (185 and 192 kDa) and the limited resolution of the SEC, the actual identity of this complex, or the possible coexistence of both species, could not be elucidated. However, an albumin dimer was observed exclusively in the 1:2 peak, hinting towards a one FabA to two albumin stoichiometry (Figure 1E).

Next, we assessed the pharmacokinetic profile in rabbits after IVT injection of FabA (Figure 2). Considering the lack of rabbit cross-reactivity of the anti-HSA nanobody (Table 3), we co-formulated FabA with HSA at HSA:FabA molar ratios of approximately 1.2 and 17.9. The study showed small differences in the ocular PK of the low albumin concentration group when compared to the group that received FabB, not bearing the anti-HSA nanobody (Table 5). Approximately a 3-fold increase in vitreal exposure and half-life was instead observed for the high albumin concentration group. These observations suggest the requirement for a certain minimum albumin concentration to allow the formation of FabA–albumin complex(es) with distinct pharmacokinetic behavior.

To characterize the nature of the complexes present in vivo, we conducted additional SEC studies with radiometric detection (Figure 3 and Figure 4). Analysis of VH samples from the rabbit study collected at 24 h post-dose showed a qualitative difference between low and high albumin concentration groups, with the former exhibiting predominantly the 1:1 complex and the latter revealing the coexistence of 1:1 and 1:2 complexes. Similarly, the formulations used in the study revealed the presence of mostly free FabA and 1:1 complex in the low albumin concentration formulation and 1:1 and 1:2 complexes for the high albumin concentration group. Overall, these findings indicate that different FabA–HSA complex profiles were injected and present in the vitreous for the two groups, resulting in differentiated vitreal PK. Furthermore, there seems to be a prerequisite for formation of the 1:2 complex to significantly reduce the clearance of FabA, consistent with the well-established principle that the hydrodynamic radius is a key determinant of ocular retention [13,16,30].

Hypothesizing that the observed half-life extension is the result of pre-formed high MW complexes, rather than their in vivo formation, we investigated the implications under physiological conditions (Figure 5 and Figure 6). We studied the formation of complexes in the human vitreous at endogenous concentrations and with supplementation of albumin. Cynomolgus monkey VH samples were also tested with the goal of identifying a more relevant animal model for potential in vivo studies. These mechanistic investigations showed very similar results in both species. Without albumin supplementation, FabA was mostly present in the free form or 1:1 complex. The amount of free FabA increased with that of FabA, suggesting that the endogenous albumin is insufficient for adequate complexation. When albumin was added to the vitreous samples, a shift towards 1:1 and eventually 1:2 complex forms was observed, with the 1:2 stoichiometry becoming predominant at concentrations of 50 µM or above. By comparison, the albumin concentrations found in the vitreous of healthy human donors (249–293 mg/mL) and cynomolgus monkeys (35–87 mg/mL) (Table 1) are one to two orders of magnitude lower. These results indicate that significantly higher than physiological concentrations of albumin are required for sufficient complexation with FabA, warranting supplementation of albumin in a potential application to cynomolgus monkeys and humans.

Additionally, we performed studies in a human vitreous sample obtained from a diabetic retinopathy patient to assess how the disease state may alter the FabA–albumin complexation profile (Figure 7). In fact, previous studies have shown that patients with retinopathies exhibit increased levels of vitreous albumin, possibly as a result of the vascular leakage associated with the disease (Table 1). In the in vivo rabbit study, the expected albumin concentration for the high albumin dose was between 10 and 15 µM, assuming a VH volume of 1.0–1.6 mL. Therefore, the albumin levels in the human diseased VH should be sufficient to form the 1:2 complex. This was, however, not reflected in the in vitro experiment where the FabA–albumin complex was found predominantly in its 1:1 conformation (Figure 7A). Our results, therefore, suggest similar complexation profiles in patients with and without eye disease and again underline the need for exogenous albumin supplementation to achieve higher order complexes.

In the dynamic in vivo setting, where the concentrations of all entities (FabA, HSA and their complexes) decrease with time after dosing, it is relevant to take into account the kinetics of formation and dissociation of the complexes. If the 1:2 complex forms and dissociates rapidly, as time passes and the concentrations of precursors decrease, it would be expected that the clearance of total FabA accelerates. In contrast, if the 1:2 complex form is irreversible or quasi-irreversible, it would determine the ocular half-life of the binder, being the lowest clearance species. The results of the in vivo rabbit study suggest that the 1:2 complex is rather stable since it was found in significant amounts in both early and late vitreous samples, without an apparent acceleration of clearance at the later time points. We tested this hypothesis by diluting pre-formed complexes in PBS and vitreous humor. However, in both conditions a dissociation of the complex was observed (data not shown). There seems to be a more elaborate underlying mechanism of complex stability than we could reproduce in vitro. Although the nature of the 1:2 complex could not be determined, it is known that albumin has a propensity to aggregate in vitro at concentrations lower than those used for the PK study formulations [31,41]. The aggregation process has been suggested to involve intermolecular disulfide bridging [31,42]. The formation of covalent bonds between two albumin molecules would be consistent with the high stability of the complex, which may be further reinforced by additional interactions with the FabA molecule.

Despite the extension of ocular half-life, a number of caveats arise when considering the potential use of albumin binding as an IVT drug delivery strategy. Co-formulation with albumin seems necessary to induce the formation of longer-residence complexes. This requirement adds on to the technical complexity of the formulation with regard to stability and other manufacturing aspects. Although the vitreal concentrations of albumin are higher in disease, the levels remain insufficient and may be expected to decrease as disease resolves by therapeutic intervention, making the approach self-limiting. Further, it is disadvantageous that binding to serum albumin has been shown to result in decreased systemic clearance and prolonged plasma half-life, after the drug diffuses out of the eye into circulation [20,43].

In this context, designing a larger molecule may be a more practical strategy to achieve the desired increase in hydrodynamic radius than introducing an albumin-binding motif. In addition, albumin is a soluble, mobile species of relatively small size and short ocular half-life. A greater reduction in diffusivity could be achieved by binding to a static retention target, such as collagen type II or hyaluronic acid, which are constituents of the vitreous matrix [20]. Whilst the compact 3D structure of collagen limits its carrying capacity to about 0.1 mg of a Fab [20], hyaluronic acid displays interesting features in terms of its vitreal abundance (34–700 μg/mL) that increases with age [44] and low turnover [45]. These characteristics yield an estimated carrying capacity of approximately 2 mg of a Fab [20], which seems suitable for ocular drug delivery.

In conclusion, we have studied the formation of complexes between albumin and a Fab fragment linked to an albumin binding nanobody and the conditions under which the ocular retention in rabbits is extended. In addition, we have addressed the human relevance of these findings, with the aim of assessing the viability as an ocular drug delivery strategy for the treatment of retinopathies. The need for supplementation of exogenous albumin in a co-formulation and the associated development hurdles led us to conclude that the approach is of limited potential relative to other delivery strategies.

## Figures and Tables

**Figure 1 pharmaceutics-12-00810-f001:**
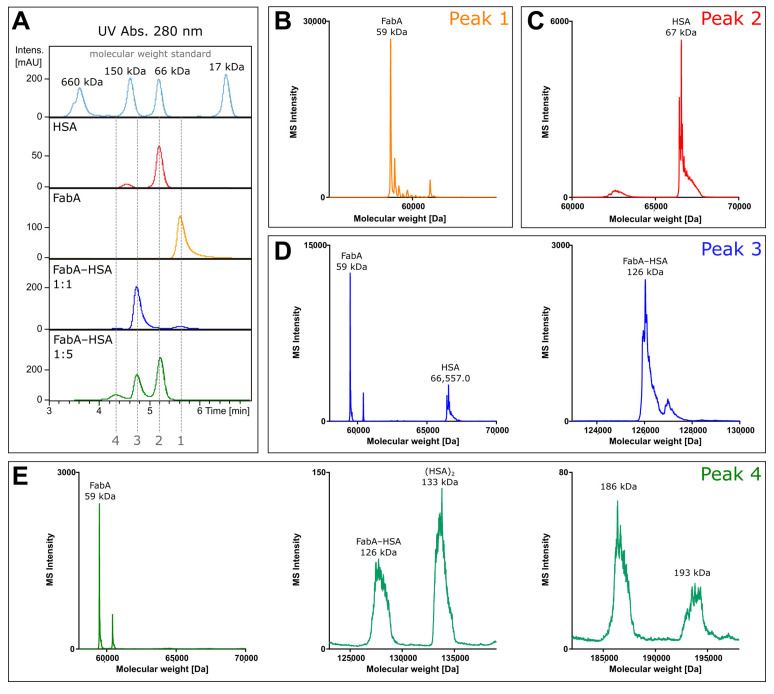
Native mass spectrometry of intact proteins. (**A**) UV size exclusion chromatograms FabA, HSA and FabA–HSA complexes formed after 4 h incubation with human VH supplemented with albumin (at 1:1 and 1:5 ratio). (**B**) Deconvoluted intact mass spectrum of chromatographic peak 1 corresponding to a molecular weight of FabA and (**C**) HSA. (**D**) Deconvoluted spectra of two mass ranges. The lower mass range contains FabA and HSA, while the higher mass range covers the equimolar FabA–HSA complex of 126 kDa. (**E**) Deconvoluted intact mass spectra of three mass ranges, with FabA in the lowest, FabA–HSA and dimeric (HSA)_2_ in the middle, and two high molecular mass complexes of 186 and 193 kDa in the highest.

**Figure 2 pharmaceutics-12-00810-f002:**
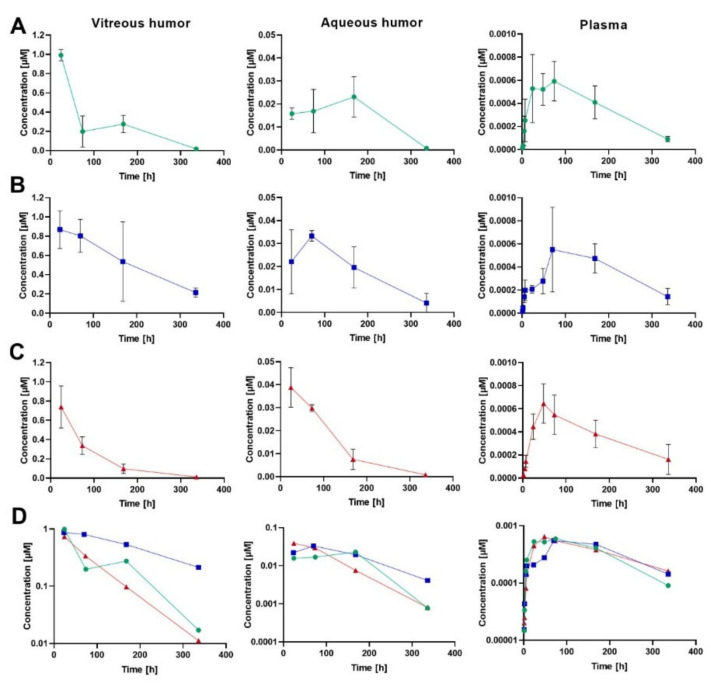
Concentration-time profiles of FabA and FabB in rabbit VH, AH and plasma after a single IVT injection of 0.05 mg per eye of the respective molecule co-formulated with HSA. Panel (**A**–**C**) represent the concentrations obtained in each matrix for group 1 (FabA and 1 nmol HSA), group 2 (FabA and 15 nmol HSA) and group 3 (FabB and 1 nmol HSA), respectively. Composite data are plotted as mean ± S.D. Panel (**D**) depicts overlaid semi-log profiles of group 1, 2 and 3 without error bars for greater readability.

**Figure 3 pharmaceutics-12-00810-f003:**
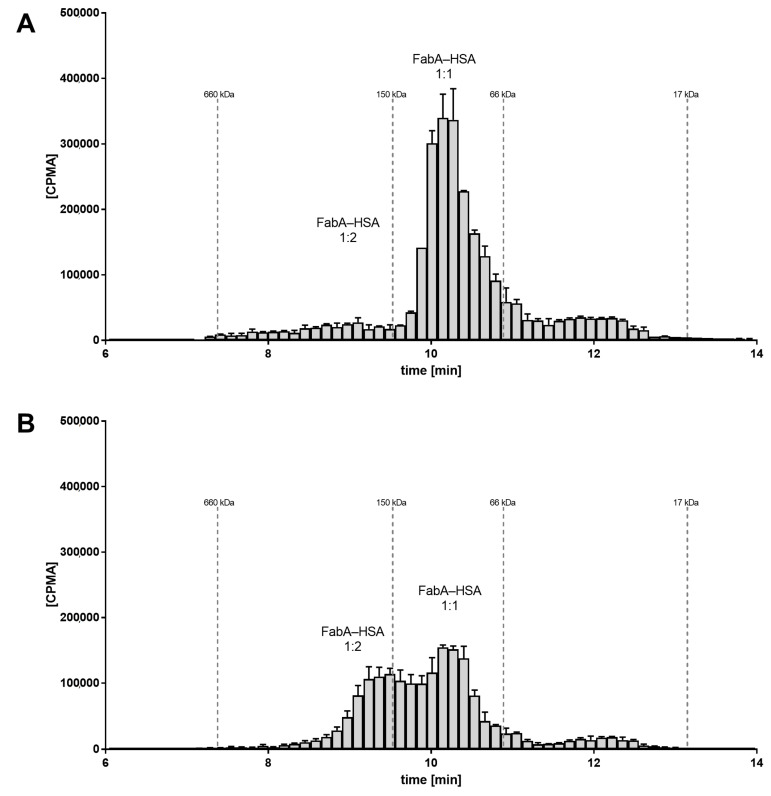
Radio-size exclusion chromatograms of FabA–HSA complexes in rabbit VH samples from the PK study (24 h time point). Error bars are min/max values of the right and left eye. Dotted lines represent the UV absorbance maxima of the MW standard. At low albumin levels (0.67 µM), the majority of the FabA–HSA complexes occur in 1:1 stoichiometry, corresponding to a MW of 126 kDa (**A**). At 15:1 HSA to FabA ratio, higher MW complexes (>150 kDa) were observed (**B**).

**Figure 4 pharmaceutics-12-00810-f004:**
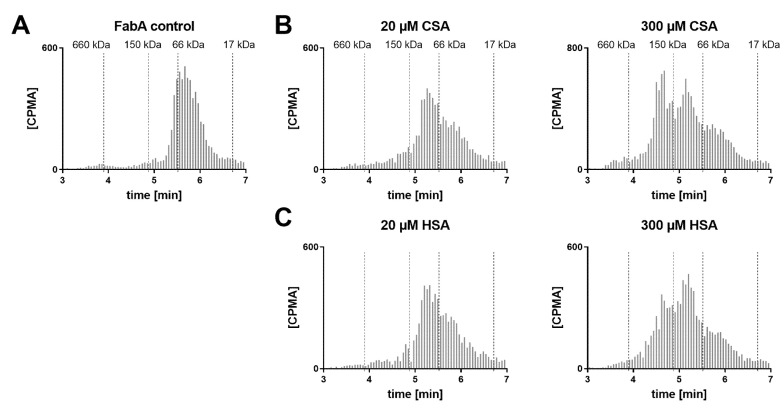
Radio-size exclusion chromatograms of dosing formulations with (**A**) FabA alone and increasing concentrations of (**B**) CSA or (**C**) HSA.

**Figure 5 pharmaceutics-12-00810-f005:**
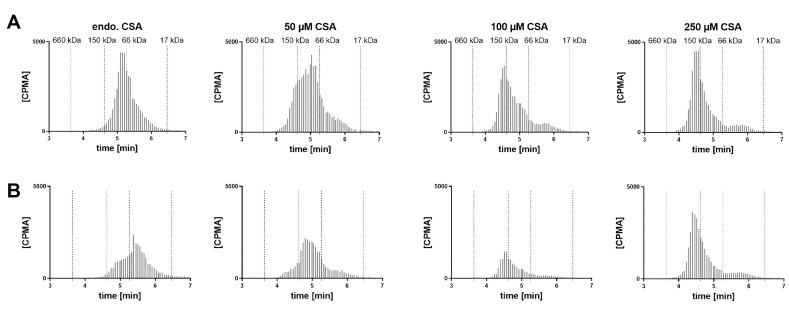
Radio-size exclusion chromatograms of FabA–CSA complexes formed after 4 h incubation in cynomolgus monkey VH supplemented with albumin. Two different concentrations of FabA were tested, namely (**A**) 0.6 and (**B**) 6 µM. Endogenous albumin concentrations were supplemented with 50, 100 and 250 µM of CSA.

**Figure 6 pharmaceutics-12-00810-f006:**
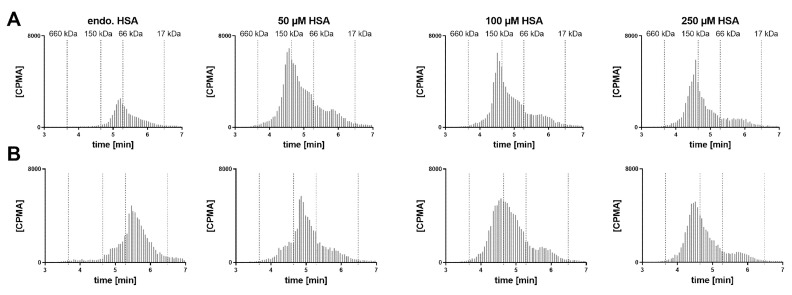
Radio-size exclusion chromatograms of FabA–HSA complexes formed after 4 h incubation in human VH and supplemented with increasing amounts of albumin. Two different concentrations of FabA were tested, namely (**A**) 0.6 and (**B**) 6 µM. Endogenous albumin concentrations were supplemented with 50, 100 and 250 µM of HSA.

**Figure 7 pharmaceutics-12-00810-f007:**
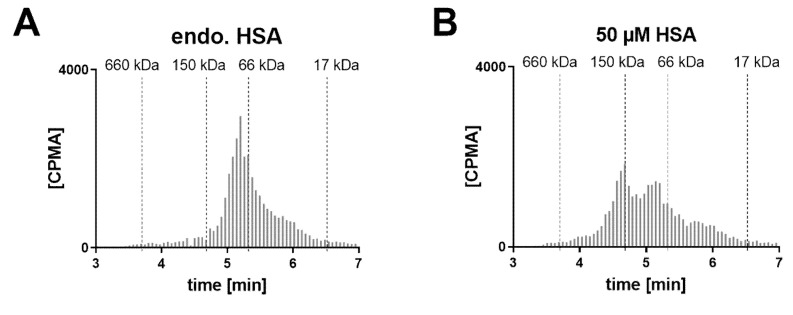
Radio-size exclusion chromatograms of FabA–HSA complexes formed after 4 h incubation with human VH (donor with eye disease) supplemented with albumin. The FabA concentration was 0.57 µM and the HSA concentration was (**A**) at endogenous levels or (**B**) at 50 µM.

**Table 1 pharmaceutics-12-00810-t001:** Summary of literature values of albumin concentration in the VH of healthy humans and cynomolgus monkeys, as well as patients undergoing vitrectomy.

Species	Disease	Reported Albumin Concentration (mg/L)	Measure ^1^	Calculated Albumin Concentration (µM) ^2^	Reference
Human	Healthy ^3^	293 ± 18	Mean ± S.E.M. (95% confidence interval: 155.8–429.4)	4.41 ± 0.27	[35,36]
		249 ± 150	Mean ± S.E.M.	3.75 ± 2.26	[35,36,37]
	Non-diabetic patients	300	Median (range: 83–1900)	4.52	[38]
	Diabetic patients without retinopathy or proliferative diabetic retinopathy (PDR)	280 ± 10	Mean ± S.D.	4.22 ± 016	Current study ^3^
		700	Median (range: 80–1200)	10.5	[38]
	Non-PDR patients	1160 ± 360	Mean ± S.E.M.	17.47 ± 5.42	[39]
	PDR	699 ± 77	Mean ± S.D.	10.5 ± 1.16	Current study ^3^
		536 ± 212	Mean ± S.E.M. (95% confidence interval: 81.3–991.1)	8.07 ± 3.19	[35,36]
		1600	Median (range: 700–3000)	24.10	[38]
		2564 ± 665	Mean ± S.E.M.	38.61 ± 10.0	[39]
	Traumatic proliferative vitreoretinopathy	743 ± 232	Mean ± S.E.M. (95% confidence interval: 217.9–1267)	11.2 ± 3.49	[35,36]
	Idiopathic proliferative vitreoretinopathy	2215 ± 867	Mean ± S.E.M. (95% confidence interval: 254.6–4176.4)	33.36 ± 13.1	[35,36]
Cynomolgus monkey	Healthy	35 ± 4	Mean ± S.D.	0.53 ± 0.05	Current study ^3^
		86.8 ± 4.0	Mean ± S.E.M. (calculated from reported 40% of total protein content: 217 µg/mL ± 4.6%)	1.31 ± 0.060	[40]

^1^ S.E.M.: standard error of the mean; S.D.: standard deviation.^2^ Calculated using albumin MW of 66.4 kDa.^3^ Vitreous samples collected post mortem.

**Table 2 pharmaceutics-12-00810-t002:** List of amounts, volumes and concentrations used in all experiments.

Experiment	FabA Dose	^3^H-FabA Concentration ^1^		Albumin Concentration ^2^	
		µM	mg/L	µM	mg/L
Formulation of in vivo rabbit study		16.8	1000	20.0	1328
		16.8	1000	300	19,920
In vivo rabbit VH	Low dose ^3^	0.57	33	0.67	44
		0.57	33	10.0	664
In vitro cynomolgus monkey VH	Low dose	0.57	33	0.53	35
		0.57	33	50.0	3320
		0.57	33	100	6640
		0.57	33	250	16,600
	High dose ^4^	5.7	330	0.53	35
		5.7	330	50.0	3320
		5.7	330	100	6640
		5.7	330	250	16,600
In vitro human VH (no eye disease)	Low dose	0.57	33	4.22	280
		0.57	33	50.0	3320
		0.57	33	100	6640
		0.57	33	250	16,600
	High dose ^4^	5.7	330	4.22	280
		5.7	330	50.0	3320
		5.7	330	100	6640
		5.7	330	250	16,600
In vitro human VH (PDR)	Low dose	0.57	33	10.53	699
		0.57	33	50.0	3320

^1^ Calculated using FabA MW of 59.5 kDa. ^2^ Calculated using albumin MW of 66.4 kDa. ^3^ Calculated assuming a rabbit vitreous humor volume of 1.5 mL [13]. ^4^ 10-fold of low dose (0.57 µM ^3^H-FabA complemented with 5.13 µM unlabeled FabA).

**Table 3 pharmaceutics-12-00810-t003:** Binding kinetics of FabA and FabB to albumin of different species.

Compound	HSA			CSA			RSA			PSA		
	ka (1/Ms)	kd (1/s)	KD (nM)	ka (1/Ms)	kd (1/s)	KD (nM)	ka (1/Ms)	kd (1/s)	KD (nM)	ka (1/Ms)	kd (1/s)	KD (nM)
FabA	110,000 ± 10,000	0.0061 ± 0.0001	55 ± 6	99,000 ± 4000	0.0065 ± 0.0004	66 ± 6	n.d.	n.d.	n.d.	n.d.	n.d.	n.d.
FabB	n.d.	n.d.	n.d.	n.d.	n.d.	n.d.	n.d.	n.d.	n.d.	n.d.	n.d.	n.d.

n.d.: no detectable interaction.

**Table 4 pharmaceutics-12-00810-t004:** Binding kinetics of FabA (unlabeled and radiolabeled) and FabB to HSA.

Compound	HSA		
	ka (1/Ms)	kd (1/s)	KD (nM)
FabA	64,400	0.00640	99
Radiolabeled FabA	64,700	0.00626	97
FabB	n.d.	n.d.	n.d.

n.d.: no detectable interaction.

**Table 5 pharmaceutics-12-00810-t005:** Pharmacokinetic parameters calculated for FabA and FabB in rabbit VH, AH and plasma after a single IVT injection of 0.05 mg of the respective molecule co-administered with 1 nmol (FabA and FabB) and 15 nmol HSA (FabA). The parameters were computed by non-compartmental analysis of composite profiles, assuming extravascular (plasma and AH) or intravenous (VH) dosing.

Parameter	Unit	Vitreous Humor			Aqueous Humor			Plasma		
		FabA 1 nmol HSA	FabA 15 nmol HSA	FabB 1 nmol HSA	FabA 1 nmol HSA	FabA 15 nmol HSA	FabB 1 nmol HSA	FabA 1 nmol HSA	FabA 15 nmol HSA	FabB 1 nmol HSA
*C* _max_	nmol/mL	0.99	0.87	0.74	0.023	0.033	0.039	0.0006	0.0006	0.0007
*t* _max_	h	24	24	24	168	74	24	72	72	48
*t* _1/2_	h	58.0	149	52.1	50.9	87.0	50.4	94.6	130	147
*MRT* _inf_	h	83.8	213	72.5	129	149	86.5	154	213	224
*AUC* _inf_	h∙nmol/mL	104	230	72.8	5.32	6.68	4.70	0.14	0.15	0.16
*V*_ss_/*F*	mL	0.68	0.79	0.87	11.6	15.8	13.0	840	1051	1128
*CL*/*F*	mL/h	0.008	0.004	0.012	0.16	0.13	0.18	6.15	5.60	5.31

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
