# Peer review of "Understanding the Half-Life Extension of Intravitreally Administered Antibodies Binding to Ocular Albumin"

_pharmaceutics, 2020, doi:10.3390/pharmaceutics12090810_

Round 1
Reviewer 1 Report
The idea of extending the half-life by binding to albumin deserves thorough clarification. After a halfway understandable introduction, the structure, contents and arrangement of the method section hardly allow to understand the order of the experiments.
- Was the expression of the nanobody?
- When did the analysis of albumin concentrations take place?
- Were the experiments in rabbits carried out in parallel to the binding assays?
- If human samples were examined, this should be differentiated in more detail in the methods section, even if only one patient was involved.
- The literature on fluctuations, post-mortem changes and the detection methods in different compartments should be fully described for albumin. Under certain circumstances, a graph (confidence interval, sample size) as opposed to a table may be considered.
Illustrations of the structure would help the introduction.
In the discussion, future approaches should be discussed more sensibly:
- Should the focus be solely on barrier disruption or bleeding induced increases in albumin concentration?
- What problems do the authors expect from an additional/simultaneous administration of albumin?
Minor points:
Measuring Albumin via ELISA is more prone to measurement errors compared to nephelometric analysis.
Is there a reason why RSA and PSA are included in Table 3?
It is not totally clear, who recruited the patents with PDR prior analysis of the measurements? Which ethics committee did approve the study?
Did the collagen matrix of the vitreous affect the formation of aggregates? Did the authors check the experiment in any medium free of HSA?
The molar rates of the ocular drugs vary considerably. Therefore, it is hard to understand why albumin should be approximately equimolar to each of those.
Author Response
Comments and Suggestions for Authors
1. The idea of extending the half-life by binding to albumin deserves thorough
clarification. After a halfway understandable introduction, the structure, contents and arrangement of the method section hardly allow to understand the order of the experiments.
The authors wish to thank Reviewer 1 for the helpful comments and suggestions. In order to improve the clarity of presentation of the experiments, the changes described below were introduced in the Methods section.
Introductory text was added to each paragraph describing the experiments to indicate their purpose and link them to the preceding and subsequent investigations.
● Line 117: “In order to determine the affinity of the Fabs to serum albumin
from different species, interaction of the compounds with HSA, CSA, RSA
and PSA was tested by surface plasmon resonance.”
● Line 134: “FabA–HSA complex composition was characterized by
high-resolution intact mass spectrometry.”
● Line 153: “An in vivo study was performed to investigate the impact of
albumin binding on ocular pharmacokinetics.”
● Line 191: “To characterize the nature of Fab–albumin complex formation in
vivo, SEC studies were performed with the vitreous humor samples obtained
in the rabbit study.”
Section 2.6 was renamed to “In vitro sample preparation” and extended for clarity.
Section 2.7 was redacted for clarity.
2. Was the expression of the nanobody?
A detailed description of the expression procedure is provided in the Materials
paragraph of the Methods section.
3. When did the analysis of albumin concentrations take place?
Endogenous albumin concentrations were measured upon receipt of the vitreous
humor samples from the commercial provider, as now mentioned in section 2.6, line 186.
4. Were the experiments in rabbits carried out in parallel to the binding assays?
The in vitro binding kinetics experiments were measured at the beginning of the
study. The in vitro complex formation experiments were performed after the rabbit ocular study, as the in vivo results guided the selection of the experimental conditions investigated in vitro.
To increase clarity on these aspects, we have revised section 2.6, describing the in vitro experiments of complex formation after the PK study of section 2.5.
5. If human samples were examined, this should be differentiated in more detail in the methods section, even if only one patient was involved.
The human and cynomolgus monkey vitreous humor samples used in our
investigations were obtained from a commercial provider, as described in the
Materials section of the Methods. To further clarify this aspect to the reader, in the revised version it is restated in section 2.6.
6. The literature on fluctuations, post-mortem changes and the detection methods in different compartments should be fully described for albumin. Under certain circumstances, a graph (confidence interval, sample size) as opposed to a table may be considered.
To the best of our knowledge, Table 1 summarizes all quantitative information
available in the literature to date regarding albumin concentrations in human and cynomolgus monkey vitreous humor, both in vivo and post mortem. For sake of clarity, this is now explicitly stated in the manuscript (line 164).
7. Illustrations of the structure would help the introduction.
An illustrative representation of the expected complexes is included in the graphical abstract. In the discussion, future approaches should be discussed more sensibly.
8. Should the focus be solely on barrier disruption or bleeding induced increases in albumin concentration?
While the vitreal concentrations of albumin are increased in disease, based on the outcomes of our investigations we conclude that the levels are insufficient to produce an extension in ocular half-life. Also, as bleeding and vascular leakage are reduced by the treatment, a concomitant decrease in vitreal albumin may be expected, making the approach self-limiting. This concept is described in the Discussion (line 451).
9. What problems do the authors expect from an additional/simultaneous administration of albumin?
This point is addressed in the Discussion:
“Despite the extension of ocular half-life, a number of caveats arise when considering the potential use of albumin binding as an IVT drug delivery strategy. Co-formulation with albumin seems necessary to induce the formation of longer-residence complexes. This requirement adds on to the technical complexity of the formulation with regard to stability and other manufacturing aspects. Although the vitreal concentrations of albumin are higher in disease, the levels remain insufficient and may be expected to decrease as disease resolves by therapeutic intervention, making the approach self-limiting. Further, it is disadvantageous that binding to serum albumin has been shown to result in decreased systemic clearance and prolonged plasma half-life, after the drug diffuses out of the eye into circulation [43,20].”
Minor points:
10. Measuring albumin via ELISA is more prone to measurement errors compared to nephelometric analysis.
The authors’ laboratories have no experience with nephelometry but utilize ELISA assays routinely and proficiently. The error of the ELISA measurements performed in this work are comparable to those of previous reports as reported in Table 1 and therefore considered adequate for the interpretation of results.
11. Is there a reason why RSA and PSA are included in Table 3?
Binding affinity measurements with serum albumin from different species commonly used in ocular research were performed to explore the suitability for an in vivo PK study (Table 3). Both rabbit (RSA) and porcine serum albumin (PSA) showed lack of cross-reactivity, requiring the use of FabA in coformulation with human serum albumin for an in vivo experiment, as was done in our rabbit study (section 2.5).
12. It is not totally clear, who recruited the patents with PDR prior analysis of the measurements? Which ethics committee did approve the study?
The human samples were obtained from a commercial provider. Please refer to the response of comment 5.
13. Did the collagen matrix of the vitreous affect the formation of aggregates? Did the authors check the experiment in any medium free of HSA?
Complex formation of FabA with albumin was tested in PBS to control for matrix
effects in Figure 4. Compared to the vitreous humor incubations, no differences in complex formation could be observed.. Furthermore, at low endogenous albumin levels, no significant multimeric complex formation was observed either (Figures 5, 6 and 7).
14. The molar rates of the ocular drugs vary considerably. Therefore, it is hard to understand why albumin should be approximately equimolar to each of those.
The hypothesis of 1:1 stoichiometry is based on the ability of one albumin binding to the single nanobody sequence in FabA. In the PK study, a FabA dose of 0.05 mg is approximately equimolar to the 1 nmol HSA that was coadministered.
Reviewer 2 Report
The abstract lacks the significant levels of the results. Rewrite with statistical analysis.
Dilution series of HSA, CSA, RSA and PSA in the range 22.2–1800 nM were injected for 180 s, dissociation was monitored for 300 s at a flow rate of 30 μL/min. what is the basis for selection of this range.
Table 1 not necessary in the manuscript. Further, there is no comparison of the present results with Table 1 was discussed in the manuscript.
In Table 2, 4 10-fold of low dose – it is 10-folds of high dose or low dose. Recheck.
The results lacks the SD values in in vivo studies.
Author Response
Comments and Suggestions for Authors
1. The abstract lacks the significant levels of the results. Rewrite with statistical analysis.
The authors would like to express their gratitude to Reviewer 2 for the constructive feedback. With regards to the statistical analysis, the sample size of the in vivo study (n=4/group, as stated in the Methods section 2.5) is arguably too small for a determination of the level of significance.
2. Dilution series of HSA, CSA, RSA and PSA in the range 22.2–1800 nM were injected for 180 s, dissociation was monitored for 300 s at a flow rate of 30 μL/min. what is the basis for selection of this range.
The protein sequence for human albumin binding nanobody incorporated in FabA was obtained from patent WO 2012/131078 A1, which also provided the binding affinity. Therefore, the authors could set the range based on the expected affinity. The following sentence was added to the method section for clarification. Line 118: “The selected albumin concentration range was based on the binding affinity of the nanobody for HSA, as reported in the original patent [33].”
3. Table 1 not necessary in the manuscript. Further, there is no comparison of the present results with Table 1 discussed in the manuscript.
The authors are of the opinion that Table 1 aids to put the present measurements of endogenous albumin concentration in healthy and diseased conditions into the context of the results previously reported in the literature. In fact, Table 1 and its contents are mentioned several times in the Results and Discussion section (lines 336, 350, 360, 414 and 422) with reference to the outcomes and conclusions of the present work.
4. In Table 2, 4 10-fold of low dose – it is 10-folds of high dose or low dose. Recheck.
Table 2 correctly indicates that the high dose is 10-fold of the low dose. The specific activity of 3H-FabA was however kept constant by complementing the low dose of labeled FabA with unlabeled FabA only. This has now been added to the Methods section 2.6 as well for improved clarity.
5. The results lack the SD values in in vivo studies.
VH, AH and plasma concentrations of the in vivo study are indeed depicted as mean ± standard deviation (S.D.) in Figure 2, while the individual data is reported in Table S1 of the Supplementary Information. The PK parameters reported in Table 5 lack the SD values because they were calculated by NCA of composite concentration-time profiles, as stated in the caption as well as in the Methods section 2.5.